# Difference in Pupillary Diameter as an Important Factor for Evaluating Amplitude of Accommodation: A Prospective Observational Study

**DOI:** 10.3390/jcm9082678

**Published:** 2020-08-18

**Authors:** Miyuki Kubota, Shunsuke Kubota, Hidenaga Kobashi, Masahiko Ayaki, Kazuno Negishi, Kazuo Tsubota

**Affiliations:** 1Department of Ophthalmology, Keio University School of Medicine, Tokyo 160-8582, Japan; shun_kubota@keio.jp (S.K.); hidenaga_kobashi@keio.jp (H.K.); mayaki@olive.ocn.ne.jp (M.A.); tsubota@z3.keio.jp (K.T.); 2Department of Ophthalmology, Shonan Keiiku Hospital, Kanagawa 252-0816, Japan; 3Graduate School of Media and Governance, Keio University, Kanagawa 252-0882, Japan; 4Department of Ophthalmology, Otake Clinic Moon View Eye Center, Kanagawa 224-0001, Japan; 5Tsubota Laboratory, Inc., Tokyo 160-0016, Japan

**Keywords:** amplitude of accommodation, pupillary diameter, presbyopia, cataract, crystalline lens, subjective refraction, axial length of the eye

## Abstract

Presbyopia is increasing globally due to aging and the widespread use of visual display terminals. Presbyopia is a decrease in the eye’s amplitude of accommodation (AA) due to loss of crystalline lens elasticity. AA differs widely among individuals. We aimed to determine the factors that cause presbyopia, other than advanced age, for early medical intervention. We examined 95 eyes of 95 healthy volunteers (33 men, 62 women) aged 22–62 years (mean: 37.22 ± 9.77 years) with a corrected visual acuity of ≥1.0 and without other eye afflictions except ametropia. Subjective refraction, AA, maximum and minimum pupillary diameters during accommodation, axial length of the eye, and crystalline lens thickness were measured. AA was measured using an auto refractometer/keratometer/tonometer/pachymeter. The difference between maximum and minimum pupillary diameters was calculated. On multiple regression analysis, age and difference in pupillary diameter were both significantly and independently associated with AA in participants aged <44 years, but not in those aged ≥45 years. Our results suggest that the difference in pupillary diameter could be an important age-independent factor for evaluating AA in healthy individuals without cataract. Thus, improving the difference in pupillary diameter values could be an early treatment target for presbyopia.

## 1. Introduction

Presbyopia is defined as a decrease in the amplitude of accommodation (AA) of the eye resulting from the loss of elasticity in the crystalline lens. The aging of society has increased the number of people with presbyopia to 1.37 billion in 2020, which is predicted to reach 1.8 billion by 2050 [1]. The number of patients with asthenopia, neck stiffness, and headache due to the non-correction or under-correction of presbyopia has also been increasing [2,3,4,5,6], likely in part because of the widespread use of visual display terminals (VDTs) [7]. Furthermore, the worldwide loss in gross domestic product (GDP) due to the non-correction or under-correction of presbyopia in individuals younger than 65 years old was reported to be $25 billion in 2011, which is equivalent to 0.037% of the GDP [1]. This attests to the gravity of this social problem.

AA gradually decreases from early childhood, with a linear decline observed from 20 to 50 years of age [8,9]. Although subjective symptoms of presbyopia are confirmed from around the age of 40 years, large individual variations are known to occur [10,11,12]. In addition to aging, which is the greatest risk factor for the decline in AA, hypermetropia [13], temperature [14,15], female sex [13,16,17], diabetes [15], alcohol intake [13], smoking [18,19], and laser-assisted in situ keratomileusis [20] have also been implicated.

Methods for presbyopia correction include reading glasses, multifocal or monovision contact lenses, and refractive surgeries. In terms of medical treatments, pirenoxine was reported to be useful in preventing progression of presbyopia [19]. There have also been reports of supplements such as Enkin^®^ (a food product containing lutein), astaxanthin [21], docosahexaenoic acid, and composite antioxidants [22] as well as thermotherapy [23] for presbyopia correction. However, no perfect treatment has yet been developed.

Pupillary diameter is an important optical factor in accommodation. However, it has not been fully evaluated in presbyopia. We hypothesized that the dynamics of pupillary diameter may be associated with the AA in addition to conventionally recognized parameters including age, sex, lens thickness, axial length, and refraction. In this study, we aimed to identify factors affecting AA in the normal population by analyzing the data of various ocular examinations.

## 2. Materials and Methods

### 2.1. Participants

In this prospective observational study, the left eyes of 95 healthy participants aged 22–62 years, with no ocular complications other than ametropia, were studied from August 2019 until January 2020. Participants with a corrected distance visual acuity (CDVA) in the logarithm of minimal angle resolution [logMAR (CDVA)] worse than 0.0 were excluded from the study. The participants were further divided according to age, with those 45 years of age and over in one group (older group) and those under 45 years of age in the other group (young group), since most people are aware of presbyopia by the age of 45 as their AA drops to 3 diopter level [10]. This study was approved by the ethics committee at Shonan Keiiku Hospital and was conducted with the written consent of all participants, in accordance with the Declaration of Helsinki.

### 2.2. Ocular Examinations

The subjective refraction, CDVA, AA, maximum and minimum pupillary diameters during accommodation, axial length of the eye, and crystalline lens thickness were measured for all participants. AA was measured using an auto refractometer/keratometer/tonometer/pachymeter (TONOREF III^®^; NIDEK, Tokyo, Japan). The maximum and minimum pupillary diameters were also measured simultaneously using the same device. The subjects were instructed to look at the internal target of TONOREF III^®^ monocularly without the aid of contact lenses or glasses. While the target was moving closer from the initial position, continuous measurement of refraction and pupil size was performed concomitantly, at up to 30 s. All measurements were obtained firstly from the right and then from the left eye. Since the measurement values of the left eye were determined to be optimal based on a preliminary study, the analysis was performed using the results of the left eye. The difference in pupillary diameter (DPD) was defined as the difference between the maximum and minimum pupillary diameters during the measurement of AA. Axial length and crystalline lens thickness were measured using the IOL Master 700^®^ (Carl Zeiss Meditec AG, Jena, Germany).

### 2.3. Measurement of AA and Pupillary Diameter by TONOREF III^®^

The subjects were instructed to look at the internal target of TONOREF III^®^ monocularly, and the objective refraction was measured first. Next, the target was moved from the initial position closer to the eye, and the continuous measurement of refraction and pupil size was performed concomitantly. At up to 30 s the accommodation amplitude (AA) was calculated automatically by subtracting the initial objective refraction (i.e., the minimum refraction) from the maximum refraction during measurement. Therefore, AA could be measured using this device regardless of the subject’s refractive error. The representative results of AA and pupillary diameters using TONOREF III^®^ are shown in Figure 1.

### 2.4. Statistical Analysis

The distributions of the continuous variables are presented as mean ± standard deviations. Student’s *t*-test was used to compare differences in normally distributed clinical parameters including age, CDVA, crystalline lens thickness, maximum pupillary diameter, and minimum pupillary diameter between the young and old groups. The Mann-Whitney U test was used to compare the non-normally distributed clinical parameters, such as subjective refraction, AA, axial length, and DPD, between the young and older groups. The chi-square test was used to compare sex between the young and older groups [24,25,26]. For these 10 factors, our null hypothesis was that there is no difference between the younger and older group. A single regression analysis was used to investigate the correlation between age and AA, DPD and AA, age and DPD. For multiple regression analysis, AA was used as the dependent variable, and age, sex, axial length, crystalline lens thickness, subjective refraction, and DPD were used as explanatory variables. Multiple regression analysis with interaction term was used to indicate whether the multiple significant factors obtained from a multivariate analysis were independent. If the significant factors were independent, the two items indicated by x are *p* ≥ 0.05 in interaction term. All analyses were conducted using SPSS version 25 for Windows (IBM Corp., Armonk, NY, USA). A *p*-value of <0.05 was considered statistically significant.

## 3. Results

### 3.1. Participant Profiles and Results of Ocular Examinations

The participant profiles and results of the ocular examinations are presented in Table 1. There were significant differences in age, AA, crystalline lens thickness, maximum pupillary diameter, minimum pupillary diameter, and DPD between the two groups. Subjective refraction, CDVA, and axial length were not significantly different between the two groups.

### 3.2. Single Regression Analyses among AA, Age, and DPD in All Participants

The results of the single regression analyses among AA, age, and DPD are presented in Figure 2. There were significant correlations between age and AA (Figure 2a; correlation coefficient: −0.771, *p* < 0.001), age and DPD (Figure 2b; correlation coefficient: −0.420, *p* < 0.001), and DPD and AA (Figure 2c; correlation coefficient: 0.634, *p* < 0.001).

### 3.3. Factors Affecting AA in All Participants

Table 2 shows the results of the multiple regression analysis in all 95 participants. We found that DPD correlated positively with AA (standardized β coefficient (Std β) = 0.365, *t* = 5.885, *p* < 0.001), while age had a significant negative correlation with AA (Std β = −0.543, *t* = −6.046, *p* < 0.001). Table 3 shows the results of the multiple regression analysis with interaction term. The results show that DPD and age did not interact with each other (*p* = 0.506); i.e., they were significant and independent factors influencing AA.

### 3.4. Factors Affecting AA in the Young and Older Groups

Table 4 shows the results of the multiple regression analysis in the 70 participants who were under 45 years of age. The results showed that DPD had a significant positive correlation with AA (Std β = 0.438, *t* = 5.246, *p* < 0.001), meaning that people with greater differences in pupil size during accommodation also have a higher amplitude of accommodation. On the other hand, age had a significant negative correlation with AA (Std β = −0.395, *t* = −3.729, *p* < 0.001). Table 5 shows the results of the interaction term. The results showed that age and DPD were significant and independent factors that influenced AA in participants under 45 years of age.

Table 6 shows the results of the multiple regression analysis in the 25 participants aged 45 years and over. The results showed that DPD had a significant positive correlation with AA (Std β = 0.589, *t* = 3.285, *p* < 0.01), indicating that a greater difference in pupil size during accommodation is present in individuals with a higher amplitude of accommodation. Unlike the results presented in Table 2, Table 3, Table 4 and Table 5, age did not have a negative correlation with AA (Std β = −0.013, *t* = −0.069, *p* = 0.946).

## 4. Discussion

Conventionally, age was thought to be the sole factor affecting AA in normal populations. In this study, we found that DPD correlated significantly with AA, and this relationship was independent of age. Therefore, we believe that DPD can be a new indicator of AA in addition to age.

The pupil has two functional roles. The first role is the light reflex, characterized by pupil constriction in response to light, which adjusts the amount of light that enters the eyeball. The second role is miosis, which is one element of the near reflex (consisting of miosis, convergence, and lens accommodation). The latency of the light reflex increases with advancing age [27]. Conversely, miotic latency associated with the near reflex does not increase with age [28,29]. Even when presbyopia has completely developed, accommodative reaction to near stimuli is maintained [29]. The pupillary sphincter, which is responsible for miosis, is controlled by the parasympathetic nervous system. Like the ciliary ring muscles that tense during accommodation, this muscle has muscarinic M3 receptors. The function of the ciliary muscles is preserved in elderly individuals [30]. It has been reported that even in cases of intraocular lens (IOL) insertion in elderly individuals, the ciliary muscle function is maintained [30]. Other reports have indicated that the force of contraction of the ciliary muscle increases until the fifth decade of life [31].

Although most people worldwide are susceptible to age-related changes of presbyopia, certain ethnicities seem to have some inherent advantages. The Moken people of southeast Asia have good visual acuity, even while diving in the dark ocean [32]. Normally, the corneal refractive power is lost underwater, and objects appear blurry. However, it appears that the Moken people can deepen their focal length by constricting their pupils for focus adjustment. Gislen et al. [33] reported that the underwater visual acuity of European children improved after they engaged in training sessions to constrict their pupils underwater. In addition, Weng et al. recently reported that the change in pupillary diameter was correlated with AA [34]. However, the number of subjects in Weng’s study was small (35 subjects), and the interaction or dependence between age and the change in pupillary diameter is not yet elucidated [34]. The current study is the first study enrolling a large number of cases to reveal that pupillary diameter is strongly correlated with AA and that individual differences are involved in presbyopia, independent of age.

Importantly, we found that age did not correlate with AA in the older group, probably because the decrease in AA had almost plateaued in this group (Figure 2a, closed square plots). In contrast, DPD strongly correlated with AA in the older group. These results suggested that DPD is a key factor for determining AA, especially in older individuals. Future research should address the treatment of presbyopia to develop a method to increase DPD, considering that an accommodative reaction to near stimuli is maintained even when presbyopia has completely developed [29], and the function of the ciliary muscles is preserved in elderly individuals [30]. Individual differences in AA are large [10,11,12], but the reason remains unknown. Furthermore, it has been reported that the use of digital devices decreases AA even in young individuals [5,35,36]. The differences in DPD might be a reasonable explanation of those cases, and this should be addressed in future studies.

Our study has some limitations. We measured AA monocularly, and we did not check convergence. DPD and AA are considered to be a part of the near reflex, and their relationship might be influenced by abnormal convergence. In addition, the number of subjects with fully developed presbyopia, i.e., over age 50, was small. These subjects warrant further investigation. Furthermore, DPD in Table 5 was not significant in the interaction term; however, it was significant on multiple regression analysis. When the main effect in a general multiple regression analysis is not significant, the interaction term cannot be subsequently examined. The results of the main effect in multiple regression analysis are adopted in covariance analysis, which is used to examine the interaction term. Table 4 shows the results of the multiple regression analysis, while Table 5 shows those of the covariance analysis. Since the statistical methods used for these results differ slightly, it is possible that the results may vary. In any case, the data in Table 4, Table 5 and Table 6 are still auxiliary to the main results in Table 2 and Table 3. In conclusion, this study revealed that DPD is an important independent factor other than age that affects AA in all age groups. Therefore, increasing DPD using various methods, such as exercises or medications, might be a new option for the treatment of presbyopia.

## Figures and Tables

**Figure 1 jcm-09-02678-f001:**
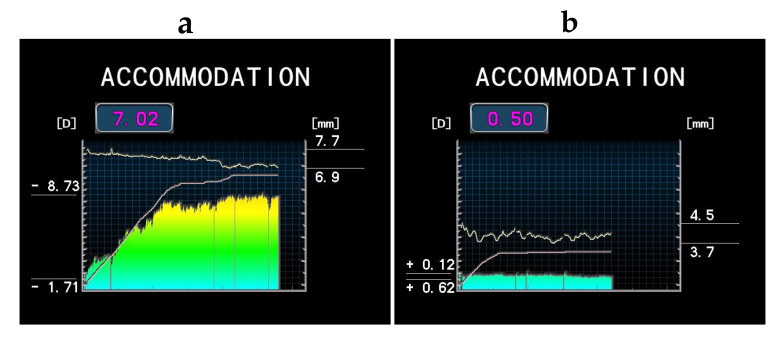
Representative results of the amplitude of accommodation (AA) and pupillary diameters measured by TONOREF III^®^; (**a**) 25-year-old female, 7.02 D of AA; (**b**) 53-year-old female, 0.50 D of AA. The *X*-axis represents an examination time of up to 30 s. The left *Y*-axis represents refraction, and the right *Y*-axis represents the pupil diameter during measurement. The upper wave is a continuously recorded pupillary diameter. The colored bars represent the real-time refraction, and the line chart represents the internal target position. The values on the left *Y*-axis are the maximum and minimum refraction values, and the magnitude calculated as AA (D) is shown in the upper square. The minimum refraction value represents refraction when the visual target is located in the initial position. The values on the right *Y*-axis are the maximum and minimum pupillary diameters.

**Figure 2 jcm-09-02678-f002:**
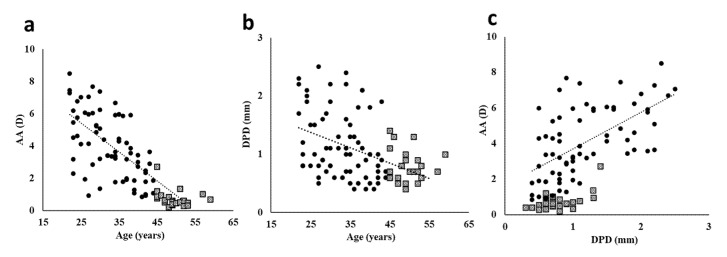
Single correlation curve of age and accommodation in 95 participants. (**a**) AA vs age (correlation coefficient, −0.771; *p* ≤ 0.01; (**b**) AA vs. DPD (correlation coefficient, −0.420; *p* ≤ 0.01); and (**c**) DPD vs. age (correlation coefficient, 0.634; *p* ≤ 0.01). AA: amplitude of accommodation, DPD: difference in pupillary diameter. ●, young group; □, older group.

**Table 1 jcm-09-02678-t001:** Participant profiles and results of ocular examinations.

	All	Young Group (<45 Years Old)	Older Group (≥45 Years Old)	*p*-Value (Young vs. Old)
Number of eyes	95	70	25	-
Age (range)	22−62	22−44	45–62	<0.01 *
Sex (male/female)	33/62	30/40	3/22	<0.01 ***
Subjective refraction (D)	−2.85 ± 2.53	−2.88 ± 2.33	−2.77 ± 3.10	0.333 **
AA (D)	3.19 ± 2.27	4.04 ± 2.00	0.74 ± 0.50	<0.01 **
CDVA	−0.10 ± 0.05	−0.11 ± 0.06	−0.10 ± 0.04	0.359 *
Axial length (mm)	24.53 ± 2.65	24.74 ± 1.23	23.94 ± 4.64	0.745 **
Crystalline lens thickness (mm)	3.93 ± 0.37	3.81 ± 0.31	4.27 ± 0.29	<0.01 *
Maximum pupillary diameter (mm)	5.65 ± 0.91	5.94 ± 0.74	4.93 ± 0.81	<0.01 *
Minimum pupillary diameter (mm)	4.58 ± 1.02	4.76 ± 1.01	4.06 ± 0.92	<0.01 *
DPD (mm)	1.07 ± 0.55	1.17 ± 0.58	0.87 ± 0.59	<0.01 **

AA, amplitude of accommodation; CDVA, corrected distance visual acuity in the logarithm of minimal angle resolution (logMAR (CDVA)); DPD, difference in pupillary diameter; *, *p*-value in Student’s *t* test; **, *p*-value in Mann-Whitney U test; ***, *p*-value in chi-square test; -, not calculated. From the test based on the null hypothesis, age, AA, crystalline lens thickness, maximum pupillary diameter, minimum pupillary diameter, and DPD were rejected since they were significant between the two groups.

**Table 2 jcm-09-02678-t002:** Multiple regression analysis in all participants (*n* = 95 and adjusted R^2^ = 0.712).

	Unstandardized		Standardized			95% CI	
	B	SE	Beta	*t*	*p*	Lower	Upper
(Constant)	7.345	2.113		3.476	0.001	3.145	11.546
Age	−0.126	0.021	−0.543	−6.046	<0.01 **	−0.168	−0.085
Sex	0.277	0.291	0.058	0.951	0.344	−0.302	0.856
Axial length	−0.058	0.052	−0.065	−1.116	0.268	−0.163	0.046
Crystalline lens thickness	0.049	0.049	0.056	0.990	0.325	−0.049	0.147
Subjective refraction	−0.672	0.536	−0.109	−1.255	0.213	−1.738	0.393
DPD	1.513	0.257	0.365	5.885	<0.01 **	1.002	2.024

Dependent variable: amplitude of accommodation. CI, confidence interval; B, partial regression coefficient; SE, standard error; DPD, difference in pupillary diameter; **, *p* < 0.01.

**Table 3 jcm-09-02678-t003:** Multiple regression analysis with interaction term (adjusted R^2^ = 0.703).

Source	Type III Sum of Squares	df	Mean Square	F	*p*
Corrected model	344.983	3	114.994	75.098	0.000
Intercept	43.455	1	43.455	28.379	0.000
Age	26.344	1	26.344	17.204	0.000
DPD	7.691	1	7.691	5.023	0.027
Age × DPD	0.682	1	0.682	0.446	0.506
Error	139.344	86	1.531		
Total	1440.115	94			

df, degree of freedom; F, the ratio of mean square for each factor to that of the error; DPD, difference in pupillary diameter.

**Table 4 jcm-09-02678-t004:** Multiple regression analysis in the young group (22–44 years old, *n* = 70 and adjusted R^2^ = 0.579).

	Unstandardized		Standardized			95% CI	
	B	SE	Beta	*t*	*p*	Lower	Upper
(Constant)	5.581	3.833		1.456	0.150	−2.080	13.242
Age	−0.120	0.032	−0.395	−3.729	<0.01 **	−0.185	−0.056
Sex	0.596	0.349	0.148	1.709	0.092	−0.101	1.293
Axial length	−0.090	0.072	−0.104	−1.246	0.217	−0.233	0.054
Crystalline lens thickness	0.174	0.137	0.106	1.271	0.209	−0.100	0.448
Subjective refraction	−1.129	0.659	−0.176	−1.714	0.091	−2.446	0.187
DPD	1.513	0.288	0.438	5.246	<0.01 **	0.937	2.090

Dependent variable: amplitude of accommodation. CI, confidence interval; B, partial regression coefficient; SE, standard error; DPD, difference in pupillary diameter; **, *p* < 0.01.

**Table 5 jcm-09-02678-t005:** Multiple regression analysis with interaction term (adjusted R^2^ = 0.523).

Source	Type III Sum of Squares	df	Mean Square	F	*p*
Corrected model	150.366a	3	50.122	26.212	0.000
Intercept	23.866	1	23.866	12.481	0.001
Age	12.427	1	12.427	6.499	0.013
DPD	1.676	1	1.676	0.876	0.353
Age × DPD	0.054	1	0.054	0.028	0.867
Error	126.203	66	1.912		
Total	1420.617	70			
Corrected total	276.570	69			

df, degree of freedom; F, the ratio of mean square for each factor to that of the error; DPD, difference in pupillary diameter. Age significantly correlated with the amplitude of accommodation, and the age variables were independent of each other.

**Table 6 jcm-09-02678-t006:** Multiple regression analysis in the older group (45–62 years old, *n* = 25 and adjusted R^2^ = 0.427).

	Unstandardized		Standardized			95% CI	
	B	SE	Beta	*t*	*p*	Lower	Upper
(Constant)	3.027	1.628		1.859	0.079	−0.394	6.447
Age	−0.001	0.021	−0.013	−0.069	0.946	−0.046	0.043
Sex	−0.483	0.265	−0.319	−1.823	0.085	−1.040	0.074
Axial length	−0.025	0.029	−0.155	−0.873	0.394	−0.087	0.036
Crystalline lens thickness	−0.008	0.017	−0.073	−0.461	0.650	−0.044	0.028
Subjective refraction	−0.581	0.380	−0.330	−1.526	0.144	−1.380	0.219
DPD	1.043	0.317	0.589	3.285	< 0.01 **	0.376	1.710

Dependent variable: amplitude of accommodation. CI, confidence interval; B, partial regression coefficient; SE, standard error; DPD, difference in pupillary diameter; **, *p* < 0.01.

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
