# Peer review of "Difference in Pupillary Diameter as an Important Factor for Evaluating Amplitude of Accommodation: A Prospective Observational Study"

_jcm, 2020, doi:10.3390/jcm9082678_

Round 1

Reviewer 1 Report

The result may be interesting and valuable, but I am not convinced in the current form of statistical analysis.

Line 58:

The letter "I" is missing at the beginning of the paragraph.

Lines 75-77

It should be better explained why "values of the left eye were determined to be
optimal". What "preliminary study" was carried out? What were its results suggesting left eyes are better for this study.

Line 95

Table 1 is difficult to read because there is no line separation. The authors use the Student's t-test without justifying the normality of the results and with one group almost 3 times larger than the other. This should be commented on in the manuscript. Furthermore, the results of the p-value are questionable, eg for the 'minimum difference in pupil diameter' between "younger" and "older" is less than 1 standard deviation, and the p-value is given as <0.01?

Line 108

There is no mark for 'older group' in figure caption

Line 112-115

The methodology of statistical analyzes has been described very briefly. For example, a description of what the generalized linear model looks like and how parameters were calculated should be added. The interpretation of the cross-impact analysis appears to be misleading. The authors show that "age" and "DPD" have a significant impact on AA (Table 2). Next, the authors analyze cross-impact of these parameters (Table 3). The results show that the impact of "age" is 3 times stronger than that of "DPD", but both relationships are significant. The authors also analyze the cross-impact of  "age" and "DPD" - this relationship turns out to be insignificant. This means that combination of DPD and age in the same person need not be related to AA.

The authors conclude that these values "do not interact (p = 0.506), ie they were significant and independent factors affecting AA". Such a statement, crucial for a manuscript, must be strongly justified. The authors only show that for patients with the DPD constant, age determines AA, for patients with constant age, DPD (weaker) determines AA, and for all patients the product of DPD and age does not determine AA. This can be justified by the strong variations of DPD values for patients of the same age, without excluding the hypothesis that for a particular patient, the age-DPD relationship explains the changes in AA due to both parameters.

Lines 125-127

The authors state that "age and DPD were significant and independent factors that influenced AA" according to Table 5. This table shows that DPD IS NOT significant factor affecting AA (p=0.353).

General remarks:

There are simple methods to find out if age is a confusing variable for DPD, such as the difference in slopes for AA (DPD) and AA (DPD, age). Cross-impact analysis is a good way to prove the opposite hypotheses (something is not related to x1 and x2, but is related to a combination of these parameters). Maybe the AA vs. (DPD x age) chart can be useful to assess the reliability of the presented conclusions?

The authors conclude that DPD may be useful in diagnosing AA in the treatment of presbyopia in the elderly, but the AA range is then so small (Fig. 1) that any differences do not appear to be very significant, and measuring DPD and AA with an optometric apparatus is equally complicated, so such an application seems of little use. Moreover, the study group does not include many patients with fully developed presbyopia, i.e aged over 50. Equalizing the number of "younger" and "older" patients could improve the reliability of the results and the correctness of the applied statistical inference.

Reviewer 2 Report

The manuscript focuses on the correlation between differences in pupillary diameter, measured during accommodation, and the amplitude of accommodation.

I have the following suggestions, for improvement:

  • Materials and Methods, participants: “Participants with a corrected distance visual acuity (CDVA) in the logarithm of minimal angle resolution [logMAR (CDVA)] lower than 0.0 were excluded”. Please replace the word “lower” for “worse”, as the CDVA is -0.10 ±
    0.05.

  • Materials and Methods, ocular examinations: The average refractive error was -2.8 D (table 1). Because participants were not using correction, and accomodation was measured using a near target, please explain how did you control for the influence of the refractive error on accomodation? Low myopia patients require less accomodative effort to see near targets. Could that have influenced the results?

Are your results extensible to the non-myopic population? Why not measure accomodation with correction?

  • Table 1 lines are not well aligned and it is difficult to ascertain which number corresponds to each item on the table.

  • Results: “The results showed that DPD had a significant positive correlation with AA”. Because these concepts are new, I suggest a more straightforward way of transmitting this information, such as “The results showed that DPD had a significant positive correlation with AA, meaning that people with larger differences in their pupil size during accommodation also have higher amplitudes of accommodation”.

  • Discussion: a final dot is missing in the sentence “The second role is miosis, which is one element of the near reflex (consisting of miosis, convergence, and lens accommodation)

Round 2

Reviewer 1 Report

Lines: 103-116

Please provide any literature references to detailed description and reliability of statistical methods used. Please also formulate null and research hypotheses for each analysis and refer to their assumptions

Lines 160-162

"Table 5 shows the results of the interaction term. The results showed that age and DPD were significant and independent factors that influenced AA in participants under 45 years of age." - thank you for your comments in the cover letter. However, it can still be read that by referring to Table 5, DPD is a significant factor influencing AA. This is not the case and this inconsistency (with Table 4) should be referred to in the text.

Lines 169-170
"Age significantly correlated with amplitude of accommodation, and age variables were independent of each other" - I think some words are missing here
